# Streaming PCA for Markovian Data

**Syamantak Kumar**[1]     **Purnamrita Sarkar**[2]

[1]Department of Computer Science, UT Austin
[2]Department of Statistics and Data Sciences, UT Austin
syamantak@utexas.edu, purna.sarkar@austin.utexas.edu

## Abstract

Since its inception in 1982, Oja's algorithm has become an established method for streaming principle component analysis (PCA). We study the problem of streaming PCA, where the data-points are sampled from an irreducible, aperiodic, and reversible Markov chain starting in stationarity. Our goal is to estimate the top eigenvector of the unknown covariance matrix of the stationary distribution. This setting has implications in scenarios where data can solely be sampled from a Markov Chain Monte Carlo (MCMC) type algorithm, and the objective is to perform inference on parameters of the stationary distribution. Most convergence guarantees for Oja's algorithm in the literature assume that the data-points are sampled IID. For data streams with Markovian dependence, one typically downsamples the data to get a "nearly" independent data stream. In this paper, we obtain the first near-optimal rate for Oja's algorithm on the entire data, where we remove the logarithmic dependence on the sample size, $n$, resulting from throwing data away in downsampling strategies.

## 1 Introduction

Streaming Principal Component Analysis (PCA) is an important and well studied problem where the principal eigenvector of the sample covariance matrix of a dataset is computed one data-point at a time. One of the most popular algorithms for streaming PCA was introduced by Erkki Oja in 1982 [29, 30]. Most existing analyses of Oja's algorithm are done when the data is sampled IID.

However, in many practical applications, the data-points are dependent and are sampled from an MCMC process converging to a target stationary distribution. This naturally arises in the context of token algorithms for Federated PCA settings [10, 12, 13] with multiple machines communicating via a fixed and connected graph topology. Each machine contains an arbitrary fraction of data-points and the goal is to design a streaming algorithm that respects this topology and returns the principal component of the whole dataset. This is typically achieved using a Metropolis-Hastings scheme that uses local information to design the transition matrix of a Markov chain with any desired stationary distribution. The stationary distribution, $\pi$, of the random walk is chosen so that the distribution of the samples under $\pi$ matches the uniform distribution over data-points. Governed by this Markov chain, a random walker then travels the network of machines and samples one data-point at a time from the current machine and computes the update. However, even under the stationary distribution, the data-points are dependent, which deviates from the IID setup. *Our goal is to obtain a near-optimal analysis of the* $\sin^2$ *error of the estimated vector with respect to the true top eigenvector of the unknown covariance matrix in the Markovian setting.*

**Estimating the first principal component with streaming PCA:**  Let $X_t$ be a mean zero $d$ dimensional vector with covariance matrix $\Sigma$, and let $\eta_t$ be a decaying learning rate. The update rule of Oja's algorithm is given as -

$$w_t \leftarrow (I + \eta_t X_t X_t^T)w_{t-1}, \ \ w_t \leftarrow \frac{w_t}{\|w_t\|_2} \tag{1}$$

where $w_t$ is the estimate of $v_1$, the top eigenvector of $\Sigma$, and $\eta_t$ is the step-size at timestep $t$. We aim to analyse the $\sin^2$ error of Oja's iterate at timestep $t$, defined as $1 - \langle w_t, v_1 \rangle^2$.

37th Conference on Neural Information Processing Systems (NeurIPS 2023).

| Paper | Markovian? | Online? | log-free main-term | $\sin^2$ error rate | Sample complexity |
|---|---|---|---|---|---|
| Jain et al. [16] | N | Y | Y | $O\left(\frac{\mathcal{V}}{\text{gap}^2}\frac{1}{n}\right)$ | $O\left(\frac{\mathcal{V}}{\text{gap}^2}\frac{1}{\epsilon}\right)$ |
| | | N | N | $O\left(\frac{\mathcal{V}\log(d)}{\text{gap}^2}\frac{1}{n}\right)$ | $O\left(\frac{\mathcal{V}\log(d)}{\text{gap}^2}\frac{1}{\epsilon}\right)$ |
| Chen et al. [3] | Y | Y | N | - | $O\left(\frac{G}{\text{gap}^2}\frac{1}{\epsilon}\log^2\left(\frac{G}{\text{gap}^2}\frac{1}{\epsilon}\right)\right)$ |
| Neeman et al. [28] | Y | N | N | $O\left(\frac{\mathcal{V}\log\left(d^{2-\frac{\pi}{4}}\right)}{(1-|\lambda_2(P)|)\,\text{gap}^2}\frac{1}{n}\right)$ | $O\left(\frac{\mathcal{V}\log\left(d^{2-\frac{\pi}{4}}\right)}{(1-|\lambda_2(P)|)\,\text{gap}^2}\frac{1}{\epsilon}\right)$ |
| **Theorem 1** | Y | Y | Y | $O\left(\frac{\mathcal{V}}{(1-|\lambda_2(P)|)\,\text{gap}^2}\frac{1}{n}\right)$ | $O\left(\frac{\mathcal{V}}{(1-|\lambda_2(P)|)\,\text{gap}^2}\frac{1}{\epsilon}\right)$ |

Table 1: Comparison of $\sin^2$ error rates and sample complexities. Here $\text{gap} := (\lambda_1 - \lambda_2)$, where $\lambda_1, \lambda_2$ are the top 2 eigenvalues of $\Sigma$ and $\mathcal{V}$ represents a suitably defined variance parameter (see assumption 2). The sample complexity represents the number of samples required to achieve $\sin^2$ error at most $\epsilon$. We note that [1] and [15] also match the online sample complexity bound provided in [16]. Further, for the offline algorithm with IID data, [17] removes the $\log(d)$ factor in exchange for a constant probability of success for large enough $n$.

**Streaming PCA in the IID setting:** For an IID data stream with $\mathbb{E}[X_i] = 0$ and $\mathbb{E}[X_i X_i^T] = \Sigma$, there has been a lot of work on determining the non-asymptotic convergence rates for Oja's algorithm and its various adaptations [16, 1, 3, 38, 14, 15, 26, 21, 25]. Amongst these, [16], [1] and [15] match the optimal offline sample complexity bound suggested by the independent and identically distributed (IID) version of Theorem 1 (See Theorem 1.1 in [16]).

We consider Oja's algorithm in the setting where the data is generated from a reversible, irreducible, and aperiodic Markov chain with stationary distribution $\pi$. We denote by $\mathbb{E}_\pi[.]$ the expectation under the stationary distribution. In this setting our goal is to estimate the principal eigenvector of $\mathbb{E}_\pi\left[X_i X_i^T\right]$. As in the IID setting, $\mathbb{E}_\pi[X_i] = 0$. The challenge is that the data, even when it reaches stationarity, is dependent. Here the degree of dependence is captured by the second eigenvalue in the magnitude of the transition matrix $P$ (denoted as $|\lambda_2(P)|$) of the Markov chain. This is closely related to the mixing time of a Markov chain [19], denoted as $\tau_{\text{mix}}$, which is the time after which the conditional distribution of a state is close in total variational distance to its stationary distribution, $\pi$ (See Section 2.1).

**Our contribution:** Using a series of approximations, we obtain an optimal error rate for the $\sin^2$ error, which is worse by a factor of $1/(1 - |\lambda_2(P)|)$ from the corresponding error rate of the IID case. Previous work [3] has established rates worse by a poly-logarithmic factor by using downsampling, i.e. applying the update on every $k^{th}$ datapoint. In Figure 1, we compare Oja's algorithm with its downsampled and offline variants (see Section 6 for more details on setup). We see that Oja's algorithm performs significantly better than the downsampled variant, and similarly to the offline variant where for the $i^{th}$ data point we compute the eigenvector of the sample covariance matrix of all data-points up-to $i$. Our work provides a concrete and novel result that explains these observations. In Table 1, we compare our bounds with related analyses of Oja's algorithm. The last row shows that we are the first to obtain an error whose main term is *free of logarithmic dependence* on $n$ or $d$ for *streaming* PCA in the *Markovian* case.

We *break the logarithmic barrier* in previous work by considering a series of approximations of finer granularity which uses reversibility of the Markov chain and standard mixing conditions of irreducible and aperiodic Markov chains. Our rates are comparable to the recent work of [28] (Proposition 1) that establishes an offline error analysis for estimating the principal component of the empirical covariance matrix of Markovian data by using a Matrix Bernstein inequality. Therefore, our results are nearly optimal in terms of the dependence on sample size, $n$, the dimension $d$, and the model parameters $\mathcal{V}, (\lambda_1 - \lambda_2)$ and $1 - |\lambda_2(P)|$ (See Section 2 for definitions). Our results also imply a linearly convergent decentralized algorithm for streaming PCA in a distributed setting. As a simple byproduct of our theoretical result, we also obtain a rate for Oja's algorithm applied on downsampled data, which is worse by a factor of $\log n$, as shown in Figure 1. To our knowledge, this is the first work that analyzes the Markovian streaming PCA problem without any downsampling that matches the error of the offline algorithm.

The crux of our analysis uses the mixing properties of the Markov chain. Strong mixing intuitively says that the conditional distribution of a state $s$ in timestep $k$ given the starting state is exponentially close to the stationary distribution of $s$, the closeness being measured using the total variation distance. All previous work on Markovian data exploits this property by conditioning on states many time steps before. However, it is crucial to a) adaptively find how far to look back and b) bound the error of the sequence of matrices we ignore between the current state and the state we are conditioning on. Observe that these two components are related. Looking back too far makes the dependence very small but increases the error resulting from approximating a larger matrix product of intermediate matrices. We present a fine analysis that balances these two parts and then uses spectral theory to bound the second part within a factor of a variance parameter that characterizes the variability of the matrices and shows up in the analysis of [16, 28].

**Related work on streaming PCA and online matrix decomposition on Markovian data:** Amongst recent work, [3] is very relevant to our setting, since it analyzes Oja's algorithm with Markovian Data samples. Inspired by the ideas of [8], the authors propose a downsampled version of Oja's algorithm to reduce dependence amongst samples and provide a Stochastic Differential Equation (SDE) based analysis to achieve a sample complexity of $O\left(\frac{G}{(\lambda_1-\lambda_2)^2}\frac{1}{\epsilon}\log^2\left(\frac{G}{(\lambda_1-\lambda_2)^2}\frac{1}{\epsilon}\right)\right)$ for $\sin^2$ error smaller than $\epsilon$, where $G$ is a variance parameter. We obtain a similar rate in Corollary 1 through our techniques. However, comparing with Theorem 1, we observe that downsampling leads to an extra $O\left(\log\left(n\right)\right)$ factor. It is important to point out that [3] provides an analysis for estimating top $k$ principal components, whereas this paper focuses on obtaining a $\log$-free error rate for the first principal component.

[22] consider the harder problem of online non-negative matrix factorization for Markovian data. Their analysis establishes asymptotic convergence of error, but does not provide a rate.

**Stochastic Optimization with Markovian Data** : Markovian models are often considered in Reinforcement Learning and Linear Dynamic Systems[2, 5, 9, 31, 4, 35, 18, 24]. There have been many notable nonasymptotic bounds for stochastic gradient descent (SGD) methods for general convex and nonconvex functions with Markovian data [8, 32, 6, 7, 10, 39, 34]. The convergence rates (sample complexities) obtained in these works apply to more general problems but do not exploit the matrix product structure inherent to Oja's algorithm. In this work, we develop novel techniques to show that a sharper analysis is possible for the PCA objective. The paper is organized as follows. Section 2 contains the problem setup and preliminaries about Markov Chains. Section 3 contains Theorem 1. We present a sketch of the main technical tools in Section 4, intermediate theorems needed for the main theorem in Section 5 and conclude with simulations in Section 6.

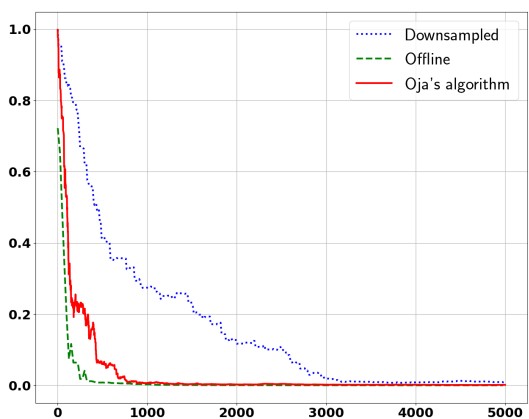

Figure 1: Comparison of Oja's algorithm with downsampled and offline variants. The X-axis represents the sample size and the Y axis represents the $\sin^2$ error of each algorithm's estimate of the leading eigenvector. The experimental setup is available in Section 6.

## 2 Problem Setup and Preliminaries

This section presents the problem setup and outlines important properties of the Markov chain that will be utilized subsequently. We assume that:

**Assumption 1.** *The Markov chain is irreducible, aperiodic, reversible, and starts in stationarity, with state distribution $\pi$*[1].

---

[1]In practice, one can let the Markov chain mix until a burn-in period of $O\left(\tau_{\mathrm{mix}}\right)$ steps to achieve state distributions close to the stationary one. A simulation experiment with different burn-in periods can be found in the Supplement Section S.6.

Such a Markov chain can arise in various situations, for e.g., while performing random walks on expander graphs which are used extensively in fields such as computer networks, error-correcting codes, and pseudorandom generators. Each state $s$ of the Markov chain is associated with a distribution $D(s)$ over $d$-dimensional vectors with mean $\mu_s \in \mathbb{R}^d$ and covariance matrix $\Sigma_s \in \mathbb{R}^{d \times d}$.

For a random walk $s_1, s_2, \cdots s_t$ on the Markov chain, we define the sequence of random variables $X_1, X_2 \cdots X_t$, where conditioned on the state $s_i$, $X_i \sim D(s_i)$. We represent the mean as $\mu := \mathbb{E}_{s \sim \pi}[\mu_s] = \sum_s \pi_s \mu_s$ and the covariance matrix as $\Sigma \in \mathbb{R}^{d \times d}$, which, for $i \in [n]$ can be expressed as:

$$\Sigma := \mathbb{E}_{s_i \sim \pi} \mathbb{E}_{D(s_i)} \left[ (X_i - \mu)(X_i - \mu)^T \right] = \mathbb{E}_{s \sim \pi}[\Sigma_s] + \mathbb{E}_{s \sim \pi}\left[ \mu_s \mu_s^T \right] - \mu \mu^T$$

In this work, we assume $\mu = 0$, which is a common assumption in the IID setting (see [16, 1]). While it may be possible to extend our analysis to the non-zero mean case, it is out of the scope of this paper. Therefore, $\Sigma = \mathbb{E}_{s_i \sim \pi} \mathbb{E}_{D(s_i)} \left[ X_i X_i^T \right]$ for $i \in [n]$

Let the eigenvalues of $\Sigma$ be denoted as $\lambda_1 > \lambda_2 \geq \lambda_3 \cdots \lambda_d$. Let $v_1$ denote the leading eigenvector of $\Sigma$ and $V_\perp$ denote the $\mathbb{R}^{d \times (d-1)}$ matrix with the remaining eigenvectors as columns. In this work, $\|.\|_2$ denotes the Euclidean $L_2$ norm for vectors and the operator norm for matrices unless otherwise specified and $I$ denotes the identity matrix of the appropriate dimensionality. We proceed under the following standard assumptions for $i \in [n]$, (see for eg. [15]).

**Assumption 2.** $\|\mathbb{E}_{s_i \sim \pi} \mathbb{E}_{D(s_i)}[(X_i X_i^T - \Sigma)^2]\|_2 \leq \mathcal{V}$.

**Assumption 3.** $\|X_i X_i^T - \Sigma\|_2 \leq \mathcal{M}$ with probability 1.

Assumption 3 also implies $\|\Sigma_s + \mu_s \mu_s^T - \Sigma\|_2 \leq \mathcal{M}$ with probability 1. WLOG, we assume $\mathcal{M} + \lambda_1 \geq 1$. We use $\mathbb{E}[.] := \mathbb{E}_{s \sim \pi} \mathbb{E}_{D(s)}[.]$ to denote the expectation over state $s \sim \pi$ and over the state-specific distributions $D(.)$, unless otherwise specified. We observe that Assumption 3 can be relaxed to accommodate sub-gaussian data distributions with appropriate variance decay (see Proposition 1 in [21]) using standard truncation-based arguments. We do not pursue this here. Define the matrix product

$$B_t := \left( I + \eta_t X_t X_t^T \right) \left( I + \eta_t X_{t-1} X_{t-1}^T \right) \ldots \left( I + \eta_1 X_1 X_1^T \right) \tag{2}$$

Unrolling the recursion in Eq 1, the output of Oja's algorithm at timestep $t$ is given as $w_t = \frac{B_t w_0}{\|B_t w_0\|_2}$.

### 2.1 Markov chain mixing times

Now we will discuss some well-known properties of an irreducible, aperiodic, and reversible Markov chain (see [19] for details and derivations). Let $|\lambda_2(P)|$ denote the second largest absolute eigenvalue of the Markov chain; let the state-distribution of the Markov chain at timestep $t$ with $s_1 = x$ be $P^t(x,.)$. For any two probability distributions $\nu_1$ and $\nu_2$, recall that the total variational distance is $TV(\nu_1, \nu_2) := \|\nu_1 - \nu_2\|_{TV} := \frac{1}{2} \sum_{x \in \Omega} |\nu_1(x) - \nu_2(x)|$. The distance from $\pi$ at the $t^{\text{th}}$ timestep is defined as $d_{\text{mix}}(t) := \sup_{x \in \Omega} TV(P^t(x,.), \pi)$. For irreducible and aperiodic Markov chains, by Theorem 4.9 in [19], we have $d_{\text{mix}}(t) \leq C \exp(-ct)$ for some $C, c > 0$. The mixing time of the Markov chain is defined as:

$$\tau_{\text{mix}}(\epsilon) := \inf\{t : d_{\text{mix}}(t) \leq \epsilon\} \tag{3}$$

As defined in Section 4.5 of [19], we will denote $\tau_{\text{mix}} := \tau_{\text{mix}}\left(\frac{1}{4}\right)$. Then, we have $\tau_{\text{mix}}(\epsilon) \leq \lceil \log_2(1/\epsilon) \rceil \tau_{\text{mix}}$. It is worth mentioning the useful relationship between $d_{\text{mix}}(.)$ and $\tau_{\text{mix}}$, given as $d_{\text{mix}}(l \tau_{\text{mix}}) \leq 2^{-l} \quad \forall l \in \mathbb{N}_0$. These results about mixing time are valid for general irreducible and aperiodic Markov chains. A reversible Markov chain satisfies $\forall \ x, y \in \Omega$, $\pi(x) P(x, y) = \pi(y) P(y, x)$. For a reversible, irreducible, and aperiodic Markov chain, the gap $1 - |\lambda_2(P)|$, is inversely proportional to (Theorem 12.4, $\tau_{\text{mix}}$ [19]).

## 3 Main Results

In this section, we present our main result, a near-optimal convergence rate for Oja's algorithm on Markovian data. As a corollary, we also establish a rate of convergence for Oja's algorithm applied

on downsampled data, where every $k^{\text{th}}$ data-point is considered. Supplement Section S.5 contains comprehensive proofs of Theorem 1 and Corollary 1 while the proof of Proposition 1 can be found in Supplement Section S.2.

**Theorem 1.** *Fix a $\delta \in (0, 1)$ and let the step-sizes be $\eta_i := \frac{\alpha}{(\lambda_1 - \lambda_2)(\beta + i)}$ with $\eta_0 \leq \frac{1}{e}, \alpha > 2$. Under assumptions 1, 2 and 3, for sufficiently large number of samples $n$ such that $\frac{n}{\log\left(\frac{1}{\eta_n}\right)} > \frac{\beta}{\log\left(\frac{1}{\eta_0}\right)}$,*

$$\beta := \frac{1000\alpha^2 \max\left\{\tau_{mix} \log\left(\frac{1}{\eta_0}\right)(\mathcal{M} + \lambda_1)^2, \frac{\left(\frac{\mathcal{V}}{1 - |\lambda_2(P)|} + \lambda_1^2\right)}{100}\right\}}{(\lambda_1 - \lambda_2)^2 \log\left(1 + \frac{\delta}{200}\right)} \tag{4}$$

*the output $w_n$ of Oja's algorithm (1) satisfies*

$$1 - \left(w_n^T v_1\right)^2 \leq \frac{C \log\left(\frac{1}{\delta}\right)}{\delta^2}\left[d\left(\frac{2\beta}{n}\right)^{2\alpha} + \frac{C_1 \mathcal{V}}{(\lambda_1 - \lambda_2)^2 (1 - |\lambda_2(P)|)}\frac{1}{n} + \frac{C_2 \mathcal{M}(\mathcal{M} + \lambda_1)^2}{(\lambda_1 - \lambda_2)^3}\frac{\tau_{mix}\left(\eta_n^2\right)^2}{n^2}\right]$$

*with probability atleast $(1 - \delta)$. Here $C$ is an absolute constant and*

$$C_1 := \frac{\alpha^2 (3 + 7|\lambda_2(P)|)}{2\alpha - 1}, \quad C_2 := \frac{35\alpha^3}{\alpha - 1}$$

**Remark 1.** *(Interpreting the $\sin^2$ error) Theorem 1 simply establishes that, under suitable choices of $\alpha, \beta$, as long as assumptions 1, 2 and 3 hold, for decaying step sizes $\eta_i = \frac{\alpha}{(\lambda_1 - \lambda_2)(\beta + i)}$, with probability at least $\frac{3}{4}$, we have,*

$$1 - \left(w_n^T v_1\right)^2 = O\left(\frac{\mathcal{V}}{(\lambda_1 - \lambda_2)^2 (1 - |\lambda_2(P)|)}\frac{1}{n}\right),$$

*for large enough $n$. This can be seen by using $\alpha = 6$, setting $\beta$ as in Eq 4, and taking $n = \tilde{\Omega}\left(d^{0.1}(\beta)^{1.2}\right)$, where $\tilde{\Omega}$ hides logarithmic factors in $n$. These choices make the first and third terms of the upper bound on the $\sin^2$ error in Theorem 1 a smaller order than the second term, which is $O(\frac{1}{n})$. For the first term, we have $d\left(\frac{2\beta}{n}\right)^{2\alpha} = O\left(\frac{1}{n^2}\right)$. The third term involves $\frac{\tau_{mix}\left(\eta_n^2\right)}{n^2}$, which is $O\left(\frac{\log^2(n)}{n^2}\right)$, using the definition of $\tau_{mix}(\epsilon)$ in Eq 3.*

**Remark 2.** *(Knowledge of mixing times) We note that setting the step-sizes $\eta_i$ in Theorem 1 requires the knowledge of the mixing time of the Markov chain, $\tau_{mix}$. An offline algorithm for estimating the mixing time is provided in [37]. They show that at least a constant $\frac{\tau_{mix}|\Omega|}{\epsilon^2}$ samples are needed for estimating $\tau_{mix}$ within an absolute error $\epsilon$. Therefore, it is possible to use the first $O\left(\frac{1}{\epsilon^2}\right)$ samples to estimate the mixing time and then set the learning rate accordingly.*

**Remark 3.** *(Regarding non-reversible Markov chains) We assume the reversibility of Markov chains to simplify our analyses. Our techniques and results extend to the non-reversible case using reversibilization tools (see [20], [11]). Specifically, one can obtain bounds in terms of the second-largest eigenvalue, in magnitude, of the modified transition matrix $P_* P$, the multiplicative reversibilization of $P$. Here, $P_*$ represents the probability transition matrix of the time-reversed chain, given as $\forall x, y \in \Omega, P_*(x, y) = \frac{\pi(y)}{\pi(x)}P(y, x)$. We leave this extension for future work.*

Next, we compare the rate of convergence proposed in Theorem 1 with the offline algorithm having access to the entire dataset $\{X_i\}_{i=1}^n$ using a recent result from [28]. Here, the authors extend the Matrix Bernstein inequality [36, 33], to Markovian random matrices. Their setup is much like ours except that the matrix at any state is fixed, i.e., there is no data distribution $D(s)$ as in our setup. However, it is easy to extend their result to our setting by observing that conditioned on the state sequence, the matrices $X_i X_i^T, i \in [n]$ are independent under our model, and we can push in the expectation over the state-specific distributions, $D(s)$, whenever required. Therefore, we have the following result -

**Proposition 1** (Theorem 2.2 of [28]+Wedin's theorem)**.** *Fix $\delta \in (0,1)$. Consider an irreducible and aperiodic Markov chain. Under assumptions 2 and 3, with probability $1 - \delta$, the leading eigenvector $\hat{v}$ of $\sum_{i=1}^{n} X_i X_i^T / n$ satisfies,*

$$1 - \left(\hat{v}^T v_1\right)^2 \le C_1' \frac{\mathcal{V} \log\left(\frac{d^{2-\frac{\pi}{4}}}{\delta}\right)}{(\lambda_1 - \lambda_2)^2} \left(\frac{1 + |\lambda_2(P)|}{1 - |\lambda_2(P)|}\right) \cdot \frac{1}{n} + C_2' \left(\frac{\mathcal{M} \log\left(\frac{d^{2-\frac{\pi}{4}}}{\delta}\right)}{(\lambda_1 - \lambda_2)(1 - |\lambda_2(P)|)}\right)^2 \cdot \frac{1}{n^2}$$

*for absolute constants $C_1'$ and $C_2'$.*

Observe that Theorem 1 matches the leading term $\frac{\mathcal{V}}{(\lambda_1 - \lambda_2)^2(1 - |\lambda_2(P)|)}$ in Theorem 1 except the $\log(d)$ term. We believe, much like the IID case (also see footnote 1 in [16]), this logarithmic term in [28]'s result is removable for large $n$ and a constant probability of success.

**Remark 4.** *(Comparison with IID algorithm) Fix a $\delta \in (0,1)$. If the data-points $\{X_i\}_{i=1}^{n}$ are sampled IID from the stationary distribution $\pi$, then under assumptions 2 and 3, using Theorem 4.1 from [16], we have that the output $w_n$ of Oja's algorithm 1 satisfies,*

$$1 - \left(w_n^T v_1\right)^2 \le \frac{C \log\left(\frac{1}{\delta}\right)}{\delta^2} \left[d \left(\frac{\beta'}{n}\right)^{2\alpha} + \frac{\alpha'^2 \mathcal{V}}{(2\alpha' - 1)(\lambda_1 - \lambda_2)^2} \frac{1}{n}\right] \tag{5}$$

The leading term of Theorem 1 is worse by a factor of $\frac{1}{1 - |\lambda_2(P)|}$. Further, it has an additive lower order term $O\left(\frac{\log^2(n)}{n^2}\right)$ due to the covariance between data-points in the Markovian case.

**Corollary 1.** *(Downsampled Oja's algorithm) Fix a $\delta \in (0,1)$. If Oja's algorithm is applied on the downsampled data-stream with every $k^{th}$ data-point, where $k := \tau_{mix}\left(\eta_n^2\right)$ then under the conditions of Theorem 1 with appropriately modified $\alpha$ and $\beta$, the output $w_n$ satisfies,*

$$1 - \left(w_n^T v_1\right)^2 \le$$
$$\frac{C \log\left(\frac{1}{\delta}\right)}{\delta^2} \left[d \left(\frac{2\beta \tau_{mix} \log(n)}{n}\right)^{2\alpha} + \frac{C_1 \mathcal{V} \tau_{mix}}{(\lambda_1 - \lambda_2)^2} \frac{\log(n)}{n} + \frac{C_2 \mathcal{M}(\mathcal{M} + \lambda_1)^2}{(\lambda_1 - \lambda_2)^3} \frac{\log^2(n) \tau_{mix}\left(\eta_n^2\right)^2}{n^2}\right]$$

*with probability atleast $(1 - \delta)$. Here $C$ is an absolute constant and $C_1 := \frac{30\alpha^2}{2\alpha - 1}$, $C_2 := \frac{35\alpha^3}{\alpha - 1}$.*

**Remark 5.** *Data downsampling to reduce dependence amongst samples has been suggested in recent work [27, 23, 3]. In Corollary 1, we establish that the bound on the rate obtained is sub-optimal compared to Theorem 1 by a $\log(n)$ factor. We prove this by a simple yet elegant observation: the downsampled data stream can be considered to be drawn from a Markov chain with transition kernel $P^k(.,.)$ since each data-point is $k$ steps away from the previous one. For sufficiently large $k$, this implies that the mixing time of this chain is $\Theta(1)$. These new parameters are used to select the modified values of $\alpha, \beta$ according to Lemma S.11 in the Supplement.*

The proof of Theorem 1 follows the same general recipe as in [16] for obtaining a bound on the $\sin^2$ error. However, the original proof techniques heavily rely on the IID setting. We carry out a refined analysis for each step under the Markovian data model by a careful control of error terms arising out of dependence. The first step involves obtaining a high-probability bound on the $\sin^2$ error, by noting that Oja's algorithm on $n$ data-points can be viewed as a single iteration of the power method on $B_n$. Therefore, fixing a $\delta \in (0,1)$ using Lemma 3.1 from [16], we have with probability at least $(1 - \delta)$,

$$\sin^2(w_n, v_1) \le \frac{C \log\left(\frac{1}{\delta}\right)}{\delta} \frac{\text{Tr}\left(V_\perp^T B_n B_n^T V_\perp\right)}{v_1^T B_n B_n^T v_1}, \tag{6}$$

where $C$ is an absolute constant. The numerator is bounded by first bounding its expectation (see Theorem 3) and then using Markov's inequality. To bound the denominator, similar to [16], we will use Chebyshev's inequality. Theorem 4 provides a lower bound for the expectation $\mathbb{E}\left[v_1^T B_n B_n^T v_1\right]$. Chebyshev's inequality also requires upper-bounding the variance of $\mathbb{E}\left[v_1^T B_n B_n^T v_1\right]$, which requires us to bound $\mathbb{E}\left[\left(v_1^T B_n B_n^T v_1\right)^2\right]$ (see Theorem 5).

# 4 Main Technical Tools

In this section, we provide a sketch of the main arguments used in our proof.

**Warm-up with downsampled Oja's algorithm:** We start with the simple downsampled Oja's algorithm to build intuition. Here, one applies Oja's update rule (Eq 1) to every $k^{th}$ data-point, for a suitably chosen $k$. For $k = \lceil L\tau_{\text{mix}} \log n \rceil$, the total variation distance between any consecutive data-points in the downsampled data stream is $O(n^{-L})$. As we show in Corollary 1, the error of this algorithm is similar to the error of Oja's algorithm applied to $n/k$ data-points in the IID setting, i.e., $O(\mathcal{V}\tau_{\text{mix}} \log n/n)$.

We will take $\mathbb{E}\left[v_1^T B_n B_n^T v_1\right]$ as an example. Let us introduce some notation.

$$B_{j,i} := \left(I + \eta_j X_j X_j^T\right)\left(I + \eta_{j-1} X_{j-1} X_{j-1}^T\right) \ldots \left(I + \eta_i X_i X_i^T\right) \tag{7}$$

We peel this quantity one matrix at a time from the inside. Note that for a reversible Markov chain, standard results imply (see Lemma 1) that the mixing conditions apply to the conditional distribution of a state given another state $k$ steps in the "future" (see Supplement Section S.3 for a proof). Recall $d_{\text{mix}}(k)$ from Section 2.1.

**Lemma 1.** *Under Assumption 1,* $\frac{1}{2} \sup_{t \in \Omega} \sum_s |\mathbb{P}\left(Z_t = s | Z_{t+k} = t\right) - \pi\left(s\right)| = d_{mix}\left(k\right)$.

It will be helpful to explain our analysis by comparing it with the IID setting. For this reason, we will use $\mathbb{E}_{\text{IID}}[.]$ to denote the expectation under the IID data model. Define $\alpha_{n,i} := \mathbb{E}\left[v_1^T B_{n,i} B_{n,i}^T v_1\right], i \in [n]$. Then we have,

$$\alpha_{n,1} = \mathbb{E}\left[v_1^T B_{n,2}\left(I + \eta_1 \Sigma + \eta_1(X_1 X_1^T - \Sigma)\right)\left(I + \eta_1 \Sigma + \eta_1(X_1 X_1^T - \Sigma)\right)^T B_{n,2}^T v_1\right]$$

$$= \mathbb{E}\left[v_1^T B_{n,2}\left(I + \eta_1 \Sigma\right)^2 B_{n,2}^T v_1\right] + 2\eta_1 T_1 + \eta_1^2 T_2, \tag{8}$$

where the first term is smaller than $(1 + \eta_1\lambda_1)^2 \alpha_{n,2}$. We define $T_1$ and $T_2$ as follows. $T_1 := \mathbb{E}\left[v_1^T B_{n,2}\left(I + \eta_1\Sigma\right)\left(X_1 X_1^T - \Sigma\right) B_{n,2}^T v_1\right]$, and $T_2 := \mathbb{E}\left[v_1^T B_{n,2}\left(X_1 X_1^T - \Sigma\right)^2 B_{n,2}^T v_1\right]$.

For the IID setting, the *second term is zero*, and the third term can be bounded as follows:

$$\mathbb{E}_{\text{IID}}\left[v_1^T B_{n,2}\left(X_1 X_1^T - \Sigma\right)^2 B_{n,2}^T v_1\right] = \mathbb{E}_{\text{IID}}\left[v_1^T B_{n,2}\mathbb{E}\left[\left(X_1 X_1^T - \Sigma\right)^2\right] B_{n,2}^T v_1\right]$$

$$\leq \mathcal{V}\mathbb{E}_{\text{IID}}\left[v_1^T B_{n,2} B_{n,2}^T v_1\right]$$

Let us denote the IID version of $\alpha_{n,i}$ by $\alpha_{n,i}^{\text{IID}} := \mathbb{E}_{\text{IID}}[v_1^T B_{n,i} B_{n,i}^T v_1]$. The final recursion for the IID case becomes: $\alpha_{n,1}^{\text{IID}} \leq (1 + 2\eta_1\lambda_1 + \eta_1^2\left(\lambda_1^2 + \mathcal{V}\right))\alpha_{n,1}^{\text{IID}}$. So, for our Markovian data model, the hope is that the cross term $T_1$ (which has a multiplicative factor of $\eta_1$) is $O(\eta_1)$ and $T_2$ is $O(\eta_1^2)$. We will start with the $T_1$ term, which is zero in the IID setting.

Figure 2: If we could replace the intermediate products (white matrices) by $I$, the conditional expectation of the noise matrix $X_1 X_1^T - \Sigma$ conditioned on the gray matrices would be nearly zero.

We will show that $T_1$, while not zero like the IID case, is still sufficiently small. Intuitively, if we could replace the $k$ matrices between $X_1 X_1^T - \Sigma$ and $B_{n,2+k}$ for some suitably large integer $k$ by identity (see Figure 2), then using (reverse) mixing properties of the Markov chain, we could argue using Lemma 1 that $\mathbb{E}[X_1 X_1^T - \Sigma | s_{1+k}, \ldots, s_n]$ is very close to zero (see Figure 2). The following lemma formally bounds the deviation of the length-$k$ matrix product from identity.

**Lemma 2.** *Let Assumption 3 hold. If* $\forall i \in [n], \eta_i k_i\left(\mathcal{M} + \lambda_1\right) \leq \epsilon, \epsilon \in (0,1)$ *and* $\eta_i$ *forms a non-increasing sequence then* $\forall\ m \leq n - k_n$,

$$\left\|B_{m+k_m-1,m} - I\right\|_2 \leq (1 + \epsilon) k_m \eta_m\left(\mathcal{M} + \lambda_1\right) \text{ and} \tag{9}$$

$$\left\|B_{m+k_m-1,m} - I - \sum_{t=m}^{m+k_m-1} \eta_t X_t X_t^T\right\|_2 \leq k_m^2 \eta_m^2\left(\mathcal{M} + \lambda_1\right)^2 \tag{10}$$

Lemma 2 bounds the norm of the matrix product $B_{t+k_t-1,t}$ at two levels. The first result provides a coarse bound, approximating linear and higher-order terms. The second result provides a finer bound, preserving the linear term and approximating quadratic and higher-order terms. The proofs involve a straightforward combinatorial expansion of $B_{t+k_t-1,t}$ and are deferred to the Supplement Section S.3.

Approximating $\prod_{i=2}^{k+1}(I + \eta_i X_i X_i^T)$ requires $\eta_1 k$ to be small. Since this is a recursive argument, we would need $\eta_i k$ to be small for $i = 1, \ldots n$, which is satisfied by the strong condition $\eta_1 k$ is small. To obtain a tight analysis, we choose $k$ adaptively. We set $k_i = \tau_{\text{mix}}(\eta_i^2)$ (see definition in Eq 3).

As we will show in detail in the Supplement, Lemma 2 Eq 10 along with the adaptive choice of $k_i$ gives us a sharp error bound. Using it, we can bound $T_1$ (see Eq 8) as:

$$T_1 \leq \sum_{j=2}^{k+1} \eta_j \mathbb{E}\left[ v_1^T B_{n,k+2} \underbrace{\mathbb{E}\left[\left(X_j X_j^T\right)\left(I + \eta_1 \Sigma\right)\left(X_1 X_1 - \Sigma\right) | X_{k+2}, \ldots, X_n\right]}_{T_{1,j}} B_{n,k+2}^T v_1 \right]$$
$$+ O(\eta_1^2 k_1^2)\alpha_{n,k+2}$$

Naively bounding the $T_{1,j}$ term by $O(1)$ leads to the same rate as downsampled Oja's algorithm.

In the following lemma, we will establish that, indeed, $T_{1,j}$ has a much smaller norm. The novelty of our bound is not just in using the mixing properties of the Markov chain but also in teasing out the variance parameter $\mathcal{V}$. We will state the lemma, in a slightly more general form as -

**Lemma 3.** *Under Assumptions 1, 2 and 3, for $i < j \leq i + k_i$,*

$$\left\| \mathbb{E}\left[\left(X_i X_i^T - \Sigma\right) S X_j X_j^T | s_{i+k_i}, \ldots s_n\right] \right\|_2 \leq \left(|\lambda_2(P)|^{j-i} \mathcal{V} + 8\eta_i^2 \mathcal{M}(\mathcal{M} + \lambda_1)\right) \|S\|_2$$

*where $k_i$ is as defined in Lemma S.11 and $S$ is a constant symmetric positive semi-definite matrix.*

Lemma 3 bounds the norm of the covariance between matrices $\left(X_i X_i^T - \Sigma\right) S$ and $X_j X_j^T$. In particular, this implies that the norm of $T_{1,j}$ decays as $|\lambda_2(P)|^{j-1}$. The proof uses a spectral argument that replaces a coarse approximation by a sum of $k_i$ $O(1)$ terms to sum of $k$ *exponentially decaying* terms, thereby removing the dependence on $k_i$, which can be as large as $\log(n)$. The proof is deferred to the Supplement Section S.4.

Let $\{c_1, c_2, c_3, c_4\}$ be positive constants for ease of notation. Coming back to Eq 8, we can bound $T_1$ as follows: $T_1 \leq \alpha_{n,k+2}\left(\eta_1 \dfrac{c_1 |\lambda_2(P)| \mathcal{V}}{1 - |\lambda_2(P)|} + c_2 \eta_1^2 k_1^2\right)$. A similar argument can be applied to bound $T_2$ as: $T_2 \leq \alpha_{n,k+2}\left(\mathcal{V} + c_3 \eta_1 k_1^2\right)$. Putting everything together in 8, we have

$$\alpha_{n,1} \leq \underbrace{\left((1 + \eta_1 \lambda_1)^2 + \mathcal{V}\right)\alpha_{n,2}}_{\text{Recursion for IID setting}} + \underbrace{\left(\dfrac{c_1 |\lambda_2(P)|}{1 - |\lambda_2(P)|}\right)\mathcal{V}\eta_1^2 \alpha_{n,k+2}}_{\text{Error due to Markovian dependence}} + \underbrace{c_4 \eta_1^3 k_1^2 \alpha_{n,k+2}}_{\text{Error due to approximation of matrix product}}$$

Recursing on this inequality gives us our bound on $\mathbb{E}\left[v_1^T B_n B_n^T v_1\right]$ (Theorem 2). We are now ready to present all our accompanying theorems.

## 5 Intermediate Theorems for Convergence Analysis

In this section, we present our accompanying theorems which are used to obtain the main result in Theorem 1. But before doing so, we will need to establish some notation. Let $k_i := \tau_{\text{mix}}\left(\eta_i^2\right)$, and the step-sizes be set as $\eta_i := \dfrac{\alpha}{(\lambda_1 - \lambda_2)(\beta + i)}$ with $\alpha, \beta$ as defined in Theorem 1. Let $\epsilon := \frac{1}{100}$. As shown in Lemma S.11 in Supplement Section S.3 our choice of step-sizes satisfy, $\forall i \in [n]$,

    **C.1** $\eta_i k_i (\mathcal{M} + \lambda_1) \leq \epsilon$         **C.2** (Slow decay) $\eta_i \leq \eta_{i-k_i} \leq (1 + 2\epsilon)\eta_i \leq 2\eta_i$

Further, we define scalar variables -

$$r := 2(1 + \epsilon)k_n \eta_n (\mathcal{M} + \lambda_1), \qquad \zeta_{k,t} := 40 k_{t+1}(\mathcal{M} + \lambda_1)^2$$
$$\psi_{k,t} := 6\mathcal{M}\left[1 + 3k_{t+1}^2(\mathcal{M} + \lambda_1)^2\right], \qquad \mathcal{V}' := \dfrac{1 + (3 + 4\epsilon)|\lambda_2(P)|}{1 - |\lambda_2(P)|}\mathcal{V} \qquad (11)$$

and recall the definitions of $B_t$ and $B_{j,i}$ in Eqs 2 and 7, respectively. We are now ready to present the theoretical results needed to prove our main result. For simplicity of notation, we present versions of the results by using $\eta_i := \frac{\alpha}{(\lambda_1 - \lambda_2)(\beta + i)}$ with $\alpha, \beta$ as defined in Theorem 1. However, these theorems are in fact valid under more general step-size schedules. We state and prove the more general versions in the Supplement Section S.4.

**Theorem 2.** *Under Assumptions 1, 2 and 3, for all $n > k_n$, and $\eta_i$ satisfying C.1 and C.2, we have,*

$$\mathbb{E}\left[v_1^T B_n B_n^T v_1\right] \le (1 + r)^2 \exp\left(\sum_{t=1}^{n-k_n} \left(2\eta_t \lambda_1 + \eta_t^2 \left(\mathcal{V}' + \lambda_1^2\right) + \eta_t^3 \psi_{k,t}\right)\right)$$

The three primary differences with the IID case are a) the $(1 + r)^2$ term, which arises since the recursion sketched in Section 4 leaves out the last $k_n$ terms which are bounded by $(1 + r)^2$; (b) the new factor of $\frac{1}{1 - |\lambda_2(P)|}$ with $\mathcal{V}$ due to the Markovian dependence between terms; and c) the extra lower order term $\eta_t^3 \psi_{k,t}$ arising from the use of Lemmas 2 and 3.

**Theorem 3.** *Let $u := \min\{t : t \in [n], t - k_t \ge 0\}$. Under Assumptions 1, 2 and 3, for all $n > u$, and $\eta_i$ satisfying C.1 and C.2,*

$$\mathbb{E}\left[\text{Tr}\left(V_\perp^T B_n B_n^T V_\perp\right)\right] \le (1 + 5\epsilon) \exp\left(\sum_{t=u+1}^{n} 2\eta_t \lambda_2 + \eta_{t-k_t}^2 \left(\mathcal{V}' + \lambda_1^2\right) + \eta_{t-k_t}^3 \psi_{k,t}\right)$$

$$\times \left(d + \sum_{t=u+1}^{n} \left(\mathcal{V}' + \eta_t \psi_{k,t}\right) C_{k,t}' \eta_{t-k_t}^2 \exp\left(\sum_{i=u+1}^{t} 2\eta_i \left(\lambda_1 - \lambda_2\right)\right)\right)$$

*where $C_{k,t}' := \left(1 + \frac{\delta}{200}\right) \exp\left(2\lambda_1 \sum_{i=1}^{u} \eta_j\right)$.*

Here, the difference is mainly in the new variable $u$, arising since the recursion stops at $u$, not 1. $(1 + 5\epsilon)$ represents the approximation of the first $u$ terms.

**Theorem 4.** *Under Assumptions 1, 2 and 3, for all $n > k_n$, $\eta_i$ satisfying C.1 and C.2, and $s := 2r + \frac{\delta}{1000}$, we have,*

$$\mathbb{E}\left[v_1^T B_n B_n^T v_1\right] \ge (1 - s) \exp\left(\sum_{t=1}^{n-k_n} 2\eta_t \lambda_1 - \sum_{t=1}^{n-k_n} 4\eta_t^2 \lambda_1^2\right)$$

This differs from its IID counterpart by a multiplicative factor of $(1 - s)$ for the same reason as before, which also makes the sums go up to $(n - k_n)$ instead of $n$. Note that for sufficiently large $n$ (Lemma S.12), $r = O\left(\frac{\log(n)}{n}\right)$ is very small and $\delta \in (0, 1)$. Therefore, $(1 - s) \approx 1$ as large $n$.

**Theorem 5.** *Under Assumptions 1, 2 and 3, for all $n > k_n$, and $\eta_i$ satisfying C.1 and C.2, we have,*

$$\mathbb{E}\left[\left(v_1^T B_n B_n^T v_1\right)^2\right] \le (1 + r)^4 \exp\left(\sum_{t=1}^{n-k_n} 4\eta_t \lambda_1 + \sum_{t=1}^{n-k_n} \eta_t^2 \zeta_{k,t}\right)$$

The differences are similar to the last theorems involving $v_1$. Surprisingly, for this, the coarse approximation suffices, leading to an absence of the $\mathcal{V}$ term in the bound. Having established these results, the final step is to substitute them into Eq 6 and follow the proof recipe described earlier. This requires significant calculations and is deferred to the Supplement Section S.5.

## 6  Experimental Validation

In this section, we present some simple experiments to validate our theoretical results. For more detailed experiments, see the Supplement Section S.6. We design a Markov chain with $|\Omega| = 10$ states, where the transition matrix entries $P_{ij}$ equal $\rho/(|\Omega| - 1)$ for $i \ne j$ and $1 - \rho$ for $i = j$. Smaller values of $\rho$ lead to larger mixing times. It can be verified that the stationary distribution $\pi = \mathcal{U}(\Omega)$ is uniform over the state-space and $|\lambda_2(P)| \approx (1 - \rho)$. We set $\rho = 0.2$ for Figures 1 and 3a, and vary it in Figure 3b. Each point in the plot is averaged over 20 random runs over different Markov chains, datasets, and initialization.

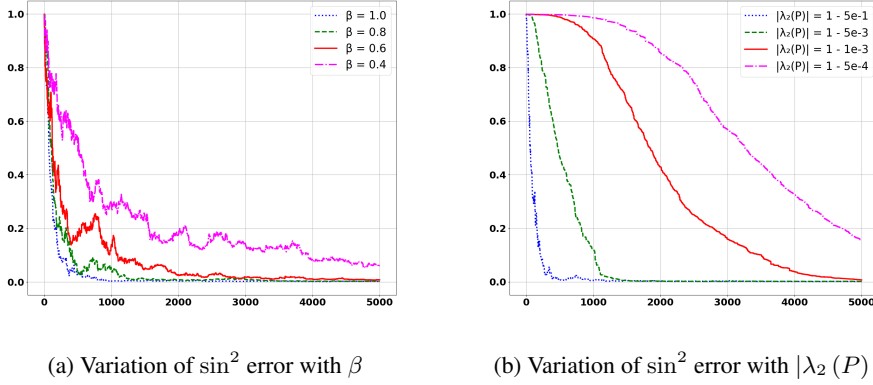

(a) Variation of $\sin^2$ error with $\beta$      (b) Variation of $\sin^2$ error with $|\lambda_2(P)|$

Figure 3: X axis represents the sample size, and Y axis represents the $\sin^2$ error.

Each state $s \in \Omega$ is associated with $D(s) := \text{Bernoulli}(p_s)$ distribution. We set $d = 1000$ and select $p_s \sim \mathcal{U}(0, 0.05)$ at the start of each random run. The covariance matrix, $\Sigma_s$, for each state is set as $\Sigma_s(i,j) = \exp(-|i-j|c_s)\sigma_i\sigma_j$ where $c_s := 1 + 9\left(\frac{s-1}{|\Omega|-1}\right), \sigma_i := 5i^{-\beta}$. We start with the stationary distribution $\pi$, and for each state $s_i$, we draw IID samples $Z_i \sim D(s_i)$. We standardize $Z_i$ such that all components have zero mean and unit variance under the state distribution, $D(s_i)$. We then generate the sample data-point for PCA as $X_i = \Sigma_i^{\frac{1}{2}} Z_i$. By construction, $\mathbb{E}_{D(s_i)}\left[X_i X_i^T\right] = \Sigma_i$ and $\mathbb{E}[X_i] = 0^d$. The step sizes for Oja's algorithm are set as $\eta_i = \frac{\alpha}{(\beta+i)(\lambda_1-\lambda_2)}$ for $\alpha = 5, \beta = \frac{5}{1-|\lambda_2(P)|}$. For the downsampled variant, every $10^{th}$ data-point is considered, and $\beta$ is accordingly divided by 10. For the offline algorithm, we recompute the leading eigenvector of the sample covariance matrix of data-points seen so far.

Figure 1 compares the performance of different algorithms for the Bernoulli distribution. Here, we are checking if the results obtained in Theorem 1, Proposition 1, and Corollary 1 are reflected in the experiments. The experimental results demonstrate that Oja's algorithm performs significantly better than the downsampled version, consistent with the theoretical results. It also shows that Oja's algorithm performs similarly to the offline algorithm, which is also confirmed by our theoretical results and that of [28]. Figure 3a compares the performance of Oja's algorithm for different covariance matrices. Smaller values of $\beta$ decrease the eigengap $\lambda_1 - \lambda_2$, and hence lead to a slower convergence. Figure 3b confirms that smaller values of $\rho$ (larger values of $|\lambda_2(P)|$) also worsen the rate, which matches with our theoretical results.

# 7 Conclusion

We have considered the problem of streaming PCA for Markovian data, which has implications in various settings like decentralized optimization, reinforcement learning, etc. The analysis of streaming algorithms in such settings has seen a renewed surge of interest in recent years. However, the dependence between data-points makes it difficult to obtain sharp bounds. We provide, to our knowledge, the first near-optimal bound for obtaining the first principal component from a Markovian data stream that breaks the logarithmic barrier present in the analysis done for downsampled data. We believe that the theoretical tools that we have developed in this paper would enable one to obtain sharp bounds for other dependent data settings, learning top $k$ principal components, and online inference algorithms with updates involving products of matrices.

# 8 Acknowledgements

We gratefully acknowledge NSF grants 2217069, 2019844, and DMS 2109155. We are also grateful to Rachel Ward, Bobby Shi, and Soumendu Sundar Mukherjee for useful discussions and to the anonymous reviewers for their valuable feedback.

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
