# OpenReview forum: "Streaming PCA for Markovian Data"
_NeurIPS.cc/2023/Conference — NeurIPS 2023 spotlight_

### Official Review · Reviewer_eXbt · 2023-06-23

**Soundness:** 3 good
**Presentation:** 2 fair
**Contribution:** 3 good
**Rating:** 6
**Confidence:** 3

**Summary:**

The submission considers the task of *streaming* PCA, estimating the largest eigenvector of a covariance matrix in an online manner as new data points arrive. The focus is on theoretical analysis of a classical method, Oja's Algorithm, in the hitherto unconsidered non-IID setting.

A seemingly strong restriction is made that the data come from a chain that is not only ergodic, but also reversible. The latter seems a potentially strong condition, but the justification appears as this is exactly the setting arising in algorithms in Federated PCA.

The primary contribution appears to be a theoretical justification for using Oja's algorithm as per normal in the considered setting, as opposed to "downsampling" points to decrease dependence between them. A multiplicative factor of how optimal error rates differ in the setting considered from the IID setting is also established.

**Strengths:**

The theoretical problem appears to be tackled in a manner involving some level of sophistication, and the results of potential interest. As the authors themselves state, many intermediate results (as contained in the supplement) may be of wider interest.

That said, whilst I am familiar with many of the individual constituent elements of this work, I am not familiar with the dependent streaming PCA literature so much, so it is difficult for me to assess the contribution.


**Weaknesses:**

* The submission is, on the whole, not particularly well written in my opinion. It could benefit from another round of two of editing and polish. The implications of *why* the particular problem are being studied and clearer discussions of its potential use would be beneficial. There are also some obvious typos and undefined notations.

* The problem considered appears quite niche and it is not made clear in the paper the extent to which the results are of wider interest (a cursory mention is made of reinforcement learning in the conclusion, but there is no discussion of this).

* As the motivation for the paper is for settings where the output comes from MCMC algorithms, I would expect an experiment exploring the effect of different burn in periods and how well Oja's method performs in practical settings and when the chain has converged to what is generally considered close enough to its stationary distribution for iterates to be used in practice (see comment below about burn in).

* As far as I can tell, simply downsampling as done in the experiment is not equivalent to the cited Chen et al. (2018) approach, who propose further refinements to the algorithm (see their Algorithm 1). As the papers primary contribution is supposedly justification for using Oja's algorithm as is over other approaches for dependent data, I would expect at a minumum a comparison is made to the actual Chen algorithm (this includes for the experiments in the supplement), as well as any other recent competitors.

* The assumption of requiring the data come from a reversible process appears somewhat strong to be of general interest. I understand the specific application, but the authors should discuss how/if this holds in a wider setting. That is, under what conditions are multivariate time series reversible? Does this hold for certain wide cases of interest?

* Issue: Page 1, footnote: It is not a fair assumption to assume the assumption holds after a sufficient period of burn in. Indeed, a burn in period may suffice to ensure that a Markov chain has burned in long enough to ensure future iterates have distribution that are sufficiently *close* to the stationary, but in general the chain never actually reaches it. Unless an analysis is provided that accounts for some initial distance to stationarity and mixing rate of the chain, I think it is a tad disingenuous to claim that the requirement for stationarity is a non-issue. I don't think not starting in stationarity is a condition that must be considered theoretically, but I **do** think at least one experiment exploring the effect of different burn ins in practical settings is required for the paper to tell a cohesive story.

**Questions:**


* The title of the submission in my opinion would benefit from being changed to being clearer about the specific setting considered. At present it masks the fact that the data are from a reversible chain, which could be quite niche.

* Suggestion: It would be clearer to say from the outset (in the abstract even) that the points are assumed to come from a reversible process in stationary.

* Why was the $\sin^2$ error used in the analysis, as opposed to say, $L_p$ norms? Is it standard in analysis of such algorithms, if so, why? The paper could benefit from discussion of this aspect.

* Issue: What are $\mathcal{V}$ and $\mathcal{M}$? These are not defined before their introduction (or even after, it seems), which is a serious problem.

* The motivation for the work is indeed Metropolis-Hastings (for example) output, which justifies the reversible setting, but how much more difficult is the problem without it? Would one expect the result to change? A small discussion of this could possibly be beneficial.

* Abstract, l2: "principle" should be "principal".

* p2, l47: "*the second eigenvalue in the magnitude of the transition matrix $P$*" reads awkwardly and incorrectly (it sounds like the magnitude is taken wrt the matrix, not the eigenvalues).

* Suggestion: Figure 1 would be easier to parse if the downsampling rate is mentioned in the legend.

* p5, l192: "atleast" should be "at least".

* p7, equation after line 250 is significantly out of the page margins.

**Limitations:**

For the most part yes, but additional clarity (as per other comments) regarding certain aspects (including the numerical experiments) would be beneficial for the paper to be up to NeurIPS standard in my opinion.

---

> ### Author Rebuttal · Authors · 2023-08-09
>
> Thank you for commenting on the novelty of our analyses and the fact that they can find applications in other online problems. We provide answers to specific queries below.
>
> 1. *Regarding wider interest*:
> Lines 19-32 mention the connection to distributed PCA where the Markov chain arising from a random walk on a connected undirected graph is naturally reversible. Furthermore, as Reviewer 1vdj pointed out, our analysis is inherently connected to the concentration of random matrix products, which has only been studied in the IID setting. We believe that the tools and results in this paper would help establish results in the Markovian case.
>
> 2. *Regarding burn-in*: Bressler et al. [25], Chen et al. [3], etc., also make the stationarity assumption. We will remove the footnote to avoid further confusion. Please see the plot in the PDF document attached with the global author rebuttal for the detailed experiment observing the effect of different burn-in periods. In general, we observed that the higher burn-in periods lead to slightly faster error decay.
>
> 3. *Regarding further refinements to Chen et al. 2018's algorithm*: We have rechecked this. The only difference we see is using the $\Pi_{\text{orth}}$ operator, which (as pointed out by Chen et al.) is the same as the orthogonalization step of Oja’s algorithm. We are also unaware of any recent competitors for streaming PCA in the Markovian setting. We would be very grateful if the reviewer points us to the refinement and other competitors they have in mind.
>
> 4. *Regarding reversibility*: Thank you for raising this important point. We assumed reversibility to simplify our analyses. However, one can extend our results to the non-reversible case following reversibilization tools (also see [A] and [B]). We use reversibility only in Lemma 3, which is then used by all other theorems. Here we look at the time-reversed chain, which is identical to the original Markov chain in the reversible case. The probability transition matrix $P_{\*}$ of the time-reversed Markov chain is given by $P_{\*}(x,y) = \frac{\pi(y)}{\pi(x)}P(y,x)$. It can also be expressed as $P_{\*}:= \Pi^{-1}P^{T}\Pi$, where $\Pi$ is a diagonal matrix with $\Pi(x,x)=\pi(x)$.  In the non-reversible case, we consider the matrix $R_{\*}:=P_{\*}P$, which is also known as the multiplicative reversibilization of $P$ (see [B]). It corresponds to the transition matrix of a reversible Markov chain and has real and nonnegative eigenvalues. In a brief sketch, we show how this allows us to replace the eigengap of the transition matrix $P$, i.e., $1-|\lambda_2(P)|$ in our current bound by $1-\sqrt{|\lambda_2(P_{\*}P)|}$.
>
>    The proof of Lemma 3 essentially reduces to computing the norm of the matrix $Q$ (Supplement line 97): $  Q:= \Pi^{\frac{1}{2}}\left(P^{t}-11^{T}\Pi\right)\Pi^{-\frac{1}{2}}. $
>    For a non-reversible Markov chain, the original transition matrix $P$ is replaced by the transition matrix $P_{\*}$ of the time-reversed Markov chain: $Q_{\*} := \Pi^{\frac{1}{2}}\left(P_{\*}^{t}-11^T\Pi\right)\Pi^{-\frac{1}{2}}$.
>
>    We will show how the norm of $Q_{\*}$ is computed in the non-reversible case. Note that $P_{\*}^t=\Pi^{-1}(P^t)^T\Pi$, and $\Pi^{\frac{1}{2}}\left(P_{\*}^{t}P^{t}\right)\Pi^{-\frac{1}{2}}$ is a symmetric matrix, with first eigenvector $\Pi^{1/2}1$.
>
>    Define: $ Q_{\*}Q_{\*}^T =
>      \Pi^{\frac{1}{2}}\left(P_{\*}^{t}P^{t} -11^{T}\Pi\right)\Pi^{-\frac{1}{2}}$. Then, we finally have,
>     $ \lVert Q_{\*}\rVert = \sqrt{\lVert Q_{\*}Q_{\*}^{T}\rVert } = \sqrt{| \lambda_{2}\left(\Pi^{1/2}P_{\*}^{t}P^{t}\Pi^{-1/2}\right)|} = \sqrt{|\lambda_{2}\left(P_{\*}^{t}P^{t}\right)|} \leq \left(\sqrt{|\lambda_{2}\left(P_{\*}P\right)|}\right)^{t},$
>    where the last inequality follows from Section 3.2 of [A].
>
> 5. *Regarding reversibility conditions for multivariate time series*: Stationary Gaussian autoregressive models are time-reversible (see [C]).
>
> 6. Questions:
>
>     Q1: *Regarding the title of the paper*:
> 	 We will add a remark stating how our argument extends to the non-reversible case.
>
>     Q2: *Regarding making the reversibility assumption clearer*: The submitted manuscript does mention the reversibility assumption in the abstract (see line 4), the introduction (see line 43), and all the theoretical results.
>
>     Q3: *Regarding the use of the $\sin^{2}$  error as opposed to $L_{p}$  norms*: The $\sin^{2}$ error is always used in the related literature since that is a standard metric to measure the distance between two unit vectors. We also want to point out that $\sin^2(v_1,\hat{v})= \frac{1}{2}\lVert v_1v_1^T-\hat{v}\hat{v}^T\rVert_F^2$.
>
>     Q4: *What are $M$ and $\mathcal{V}$?*
> 	 $\mathcal{V}$ and $M$ are defined in Assumptions 2 (line 137) and 3 (line 138), respectively.
>
>     Q5: *Regarding reversibility*: Please see our discussion to *point 4* about the extension to the non-reversible case.
>
>     Q6-10: Thank you for pointing out these typographical issues. We will update our manuscript to reflect these changes.
>
> References
>
> [A]. Lezaud, Pascal. Chernoff-type bound for finite Markov chains. Annals of Applied Probability (1998): 849-867.
>
> [B]. J. A. Fill. Eigenvalue bounds on convergence to stationarity for non-reversible Markov chains,
> with an application to the exclusion process. The Annals of applied probability pages 62–87,
> 1991.
>
> [C].  A. J. Lawrance, Directionality and Reversibility in Time Series, International Statistical Review / Revue Internationale de Statistique, Vol. 59, No. 1 (Apr., 1991), pp. 67-79

---

> > ### Comment · Reviewer_eXbt · 2023-08-10
> > **Response to Rebuttal**
> >
> > I thank the authors for their detailed rebuttal to the review and in particular for their detailed responses. The addition to the final paper of some comments regarding weaking the reversibility assumption as mentioned in the rebuttal(s) and the new numerical results, combined with seeing the responses to the other reviewers is sufficient for me to raise my score to "Weak Accept", which I have now done. The primary reason the score is not higher is largely due to the overall exposition which in my opinion, at least in its present form, detracts somewhat from what appears to be otherwise excellent work in the paper.
> >
> > Thanks in particular for clarification regarding some confusing aspects. In particular,
> >
> > * Regarding Q4 ($\mathcal{M}$ and $\mathcal{V}$), I see now that these quantities are simply constants in the assumptions .  This is simply a matter of clarity of presentation and not as serious as originally thought.
> > * Regarding Chen's 2018 algorithm - it is good to see that it is indeed equivalent as this was not immediately clear to me (though admittedly, I am not an expert in this area), perhaps a small comment can be added regarding the equivalence to avoid confusion.
> >
> > The clarification above points also contributed to my raising my score.
> >
> > Some points still outstanding albeit minor points that may benefit the final version of the paper:
> >
> > * Regarding which processes are reversible. It may be worth mentioning the reference given in the review of the paper, just to highlight that the setting is potentially of wider interest beyond the very-niche Federated PCA algorithms.
> > * Regarding Q2: Indeed, reversibility is mentioned in the abstract, but the intended point of my original comment was that the assumption that the chain is assumed to be in its  **stationarity regime** should be mentioned in the abstract.
> > * Regarding stationarity assumption in other papers. Indeed, the assumption is fine in my opinion and standard, as is to simply assume it holds after some suitably-long burn-in period. However, one should be careful saying the chain has converged or is in stationarity after burn-in or any similar statements, as this is technically not true in general. In any case, the point is minor.

---

> > > ### Author Response · Authors · 2023-08-11
> > >
> > > Dear Reviewer eXbt,
> > >
> > > We are very grateful for your consideration and kind words. We will include more references regarding reversible processes. We will also highlight the stationarity assumption in the abstract and include a discussion about the effect of burn-in in practical settings.

---

### Official Review · Reviewer_iipj · 2023-07-02

**Soundness:** 4 excellent
**Presentation:** 1 poor
**Contribution:** 3 good
**Rating:** 7
**Confidence:** 4

**Summary:**

This paper studies the problem of streaming PCA, where the data points are revealed to the algorithm one at a time and the goal is to estimate the top eigenvector of the underlying covariance matrix.
Crucially, the data points are sampled from a Markov chain and thus do not satisfy the typical assumption of being independent (they are still identically distributed as the Markov chain is stationary).
The paper analyzes the Oja's algorithm, which has shown a great empirical (and theoretical) success in streaming PCA with i.i.d. data.
Crucially, the paper does not do any downsampling of the data, which is a common practice in the literature to curb the influence of dependence in the data.

----
# Update
I thank the authors for their detailed responses. Based on their promise to include a nuanced discussion on the results in the final version of the paper (along with other presentation improvements suggested by the reviewers), I have now increased my score to 7 from 5.

**Strengths:**

The paper studies an important problem and the paper's results are novel. In particular, they seem to improve over the de-facto approach of downsampling. I like the qualitative aspects of the paper's results.


**Weaknesses:**

In terms of weaknesses, I have the following comments:

1. (Writing) The paper's writing is sadly not up to the mark. In addition to the typos (some of which are listed below), theorem statements need to be simplified and the paper (especially Sections 4 and 5) need major re-writing. Even after spending a lot of time reading through the technical proof sketches in Sections 4 and 5, I was a bit clueless.

2. (Gap-free bounds) There are no results for the gap-free setting, which is needed in several settings. I recommend adding them to strengthen the paper.

3. (Dependence on failure probability) At several places, the paper claims to match the existing bounds in the literature for the i.i.d. setting. However, the dependence on the failure probability in the paper is much worse, polynomial as opposed to logarithmic in [15].

4. (Non-zero mean setting) Can the results be generalized to the zero mean setting? In the iid setting, one can simply take a pair of data and take their difference as the sample. Can one do the same trick here in the streaming setting (without compromising the mixing time)?

**Questions:**

# Questions

1. Why are the error rates in the paper claimed to be sharp? Although it is reasonable to expect that a dependence on the mixing time appears, but does it have to be a multiplicative dependence? Are there lower bounds?

2. Does the algorithm need to know the mixing time of the chain to declare victory? That is, Theorem 1 requires n to be larger than the mixing time. If yes, is it inherent?

3. Isn't Corollary 1 immediate from the the i.i.d. result by using downsampling argument (without using Theorem 1)?

# Other Comments

1. Theorem 1 is too dense and difficult to parse.

   i. Line 158 says that the gap $1- \lambda_2(P)$ is inversely proportional to $\tau_\mathrm{mix}$. Then, why does Theorem 1 use both of them in the statement? It would be simple to use $\tau_\mathrm{mix}$.

   ii. The notation with $\alpha$ and $\beta$ is convoluted. For the version in the main theorem, I suggest choosing a good enough value of $\alpha$ and present a simplified choice of $\beta$.

2.  Line 194: Only the upper bounds obtained are sub-optimal. No lower bound has been shown.

3. Section 5: What is the connection between the series of theorem statements.


4. There are many typos and grammatical errors in the paper and many articles are missing. Several mathematical quantities are described after they are used. For example,
    i. Line 31:  w.r.t.

    ii. Line 31:  the true

    iii. Line 36: v_1 is not defined.

    iv. Line 75: What is meaning of 'stationary distribution of s

    v. Assumption 1: What exactly is the last assumption is ambiguous.

    vi. What norms are used in Assumptions 2 and 3?

    vii. Line 139: WLOG

    viii. Line 140: Why is it without loss of generality?

    vix. Line 142: Braces around the equation are missing.

    x. Section 2.1: Give specific references to the theorem statements in the book [18].

    xi. Line 181: Eq 1 should be Theorem 1.

    xii.Line 182: Which remark in [15]?

    xiii. Line 227 and 230: $\alpha_{n,2}$ is not defined.

    xiv. Line 228: $\beta_{n,2}$ is not defined.


**Limitations:**

What are some instances where $\mathcal{V}$ is much smaller than $\mathcal{M}$. Using some simple calculations, I am getting similar expressions for them in the case of Gaussian data.

---

> ### Author Rebuttal · Authors · 2023-08-10
>
> Thank you for your kind words regarding our results' novelty and the importance of the problem.
>
> 1. *Regarding writing*: We will correct the typographical errors and incorporate the suggestions. We will also add a simplified version of Theorem 1 with reasonable values substituted for the parameters.
>
> 2. *Regarding Gap-free bounds*: To our knowledge, this is the first sharp rate for Oja's algorithm for the setting with an eigengap. The gap-free bounds, although important in various scenarios, are beyond the scope of this work.
>
> 3. *Regarding dependence on failure probability*: We are comparing against Theorem 4.1 from Jain et al. [15], which has an **identical dependence** on the failure probability as us, i.e., $\frac{\log(\frac{1}{\delta})}{\delta^2}$. Hence our dependence on the failure probability *exactly matches* the existing results in the IID case.
>
> 4. *Regarding the non-zero mean case*: Thank you for raising this insightful point. Indeed, in the Markovian setting, this construction leads to a modified Markov chain with the same mixing time. Let the modified mean-zero dataset be $\\{Y_i , i = 1,\dots,\lfloor \frac{n}{2} \rfloor\\}$ where $Y_{i} := X_{2i-1} - X_{2i}$. This data comes from another Markov chain with states $(a,b)$ such that $P(a,b)>0$ in the original Markov chain. The new transition matrix $P'$ is $P'((a,b),(c,d))=P(b,c)P(c,d).$ When the original Markov chain is irreducible and aperiodic so is this one. The stationary distribution of $P'$ is $\pi'((a,b))=\pi(a)P(a,b)$. Finally, to see the relationship between the mixing times, observe that, $P(s'_t=(c,d)|s'_0=(a,b)) = P(s_t=c|s_0=b)P(c,d)$.
>
>    Thus,
>
>     $\sup_{(a,b)}\sum_{(c,d)}|P(s'_t=(c,d)|s'_0=(a,b))-\pi(c)P(c,d)|$
>
>     $=\sup_{b}\sum_c|P(s_t=c|s_0=b)-\pi(c)|$.
>
>     Therefore, the mixing time stays unchanged. Thus our analysis applies to non-zero mean distributions similar to the IID case.
>
> 5. Questions
>
>     Q1: *Regarding the sharpness of our results*: Neeman et al. [26] obtain an upper bound for offline estimation of the principal eigenvector of the empirical covariance matrix of a Markov chain. Their bound is claimed to be optimal up to logarithmic factors. Proposition 1 (line 177) provides this bound in our setting. Our error bound for streaming PCA improves this $\log(d)$ factor. Thus, in the same sense that Jain et al.’s result in [15] is sharp, we have claimed ours to be sharp. The multiplicative factor of $\tau_{\text{mix}}$ also arises in lower bounds for various convex optimization problems with Markovian data (see [8, 25]).
>
>     Q2:  *Regarding knowledge of the mixing time*:  $\tau_{\text{mix}}$ is needed to set the learning rate optimally. However, [A] provides an offline algorithm for estimating the mixing time of Markov chains. They show that at least a constant times  $\frac{\tau_{\text{mix}}|\Omega|}{\epsilon^{2}}$ samples are need for estimating $\tau_{\text{mix}}$ within an absolute error $\epsilon$ where $|\Omega|$ denotes the state space size. Therefore, one may use $O\left(\frac{1}{\epsilon^{2}}\right)$ samples to estimate the mixing time within accuracy $\epsilon$. We will add this as a remark.
>
>     Q3: *Regarding Corollary 1 being immediate from the i.i.d. result*: Intuitively, this should be the case. However, even after downsampling, the data points are not completely independent, so IID results do not apply immediately. For example, the data drop algorithm in [25] for online linear regression for Markovian data also requires a significant amount of analysis to handle dependence.
>
> 6. Other Comments
>
>    (a) *Regarding Theorem 1 presentation* :
>    Theorem 1 is presented in the most general form for easy comparison with the IID version in [25]. We will add a version including the suggestions.
>
>     *Regarding $\tau_{\text{mix}}$ and $1 - |\lambda_2\left(P\right)|$ in Theorem 1*: The relationship between $\tau_{\text{mix}}$ and $1 - |\lambda_2\left(P\right)|$ for reversible Markov chains (Eq~S.58, supplement line390) gives rise to an additional factor logarithmic in the size of the state-space, which is why we used both.
>
>    (b) *Regarding sub-optimality shown only for upper bounds (Line 194)*: We will clarify this in the final manuscript.
>
>    (c) *Regarding the connection between theorem statements (Section 5)*: Lines 200-210 show how the results in Section 5 help in proving the main result.
>
>    (d) Clarifications:
>
>    (iii) *(Line 36) Regarding $v_{1}$ not being defined*: $v_1$ is defined in line 37, which will be moved earlier.
>
>    (iv)  *(Line 75) Regarding the meaning of stationary distribution of $s$*: Stationary distribution of $s$ means $\pi(s)$.
>
>    (v) *Regarding the last assumption being ambiguous in Assumption 1*:  We mean the requirement for starting at stationarity in Assumption 1.
>
>    (vi) *Regarding norms used in Assumptions 2 and 3*: A in lines 143 and 144, $\lVert.\rVert_2$ denotes the Euclidean norm for vectors and the operator norm for matrices.
>
>    (viii) *(Line 140) Why is it without loss of generality*: We assumed $M+\lambda_1\geq 1$ because when it is less than 1, it only makes our bounds tighter.
>
>    (xii) *(Line 182) Which remark in [15]?*: We made an error in calling this a remark while it is footnote 1 on page 2.
>
>    (xiii, xiv) *(Lines 227, 228, and 230)*: $B_{n,i}$ is defined on Line 217 (Eq 7). $\alpha_{n,i}$ is defined as $\alpha_{n,i} := \mathbb{E}\left[v_1^TB_{n,i}B_{n,i}^Tv_1\right]$. PDF document in the global author rebuttal also contains a pictorial representation of $\alpha_{n,2}$.  We could not find a $\beta_{n,2}$ notation anywhere.
>
> 7. Limitations:
>
>    For Bernoulli random variables with small p, $\mathcal{V}$ is much smaller than $M^2$. We show this for a one-dimensional $Y\sim Bernoulli(p)$ as a proof of concept. Let $X=Y-p$. So $|X^2-p(1-p)|\leq 1$, i.e. $M=1$ and $\mathbb{E}(X^2-p(1-p))^2=O(p)$ for small $p$.
>
> References
>
> [A]. Wolfer, Geoffrey, and Aryeh Kontorovich. Estimating the mixing time of ergodic Markov chains. COLT 2019.

---

> > ### Comment · Reviewer_iipj · 2023-08-14
> >
> > I thank the authors for their detailed and thorough response.
> >
> > My main remaining concern is regarding the tightness/sharpness of the results: It is not clear from reading the paper what aspects of the bounds are tight (matches information-theoretic lower bounds), sharp (it is unclear what this means, anyway), or artifact of the algorithm (tight for this algorithm, but potentially improvable with some other algorithm). I believe the paper would benefit with a thorough discussion on this topic.
> >
> > For example, regarding the dependence on the failure probability: Lines 39-41 state that "*Amongst these, [15], [1] and [14] match the optimal offline sample complexity bound, suggested by the independent and identically distributed (IID) version of Theorem 1 (See Theorem 1.1 in [15]).*"
> > However, as the authors have mentioned, there is a large gap in the dependence on the failure probability in [15] vs. Matrix Bernstein. Even for Oja's algorithm, are there improved rates in the follow-up papers [1,14]? Theorem 2.3 in [14] seems to have a logarithmic dependence on the failure probability.
> >
> > Additionally, *the multiplicative factor of also arises in lower bounds for various convex optimization problems with Markovian data (see [8, 25]).*: Is there a formal lower bound that can be proved using these results?
> >
> > Finally, please add explicitly in the paper that the algorithm requires knowledge of the mixing time.

---

> > > ### Author Response · Authors · 2023-08-14
> > >
> > > *Regarding discussions on sharpness*: Thank you for the clarification. We will rephrase the word "sharp" by stating that our analysis is near-optimal in the sense that it matches the Matrix Bernstein bound for the offline PCA algorithm in Neeman et al. [26] in terms of dependence on $n$, $d$, and the model parameters $\mathcal{V}$, $(\lambda_1-\lambda_2)^2$, and $1-\lambda_2(P)$. This is the same characterization of "near-optimality" used by Jain et al. [15]. We will add a discussion to clarify this point.
> > >
> > > *Regarding failure probability*:  You are correct that Theorem 2.3 in [14] has a better dependence on the failure probability than [15]. However, [14] builds on concentration inequalities for products of IID matrices ([B]), which are not yet derived for Markovian data. We will clarify that, conceivably, the dependence on the failure probability (which matches that in [15]) can be improved with concentration inequalities on dependent matrix products.
> > >
> > > *Regarding lower bounds*: We are unaware of explicit lower bounds in the Markovian case for PCA.
> > >
> > > Both the matrix concentration and lower bounds for PCA in the Markovian case are open problems and part of our ongoing work. We will add this to the conclusion of the manuscript.
> > >
> > > *Regarding knowledge of mixing time*: We will clarify that, to set the learning rate optimally, one would need to know the mixing time. We will also discuss current work on estimating this quantity.
> > >
> > > [B] Huang D, Niles-Weed J, Tropp JA, Ward R. Matrix concentration for products. Foundations of Computational Mathematics. 2022 Dec;22(6):1767-99.

---

> > > > ### Comment · Reviewer_iipj · 2023-08-14
> > > >
> > > > I thank the authors for the clarification. Based on their promise to include a nuanced discussion on the results in the final version of the paper (along with other presentation improvements suggested by the reviewers), I have now increased my score to 7.

---

### Official Review · Reviewer_1vdj · 2023-07-04

**Soundness:** 3 good
**Presentation:** 3 good
**Contribution:** 3 good
**Rating:** 8
**Confidence:** 4

**Summary:**

This paper focuses on a particular algorithm for PCA known as the Oja's algorithm.
This is an iterative algorithm that operates in a streaming manner.
The goal is to, when the data stream is Markovian, estimate the principal component of the covariance matrix of the unknown covariate vectors.
The main question that the authors addressed is to obtain sharp (up to multiplicative constant) rate guarantee for such an algorithm.
There are miscellaneous results in other parts of the paper that compare favorably to the main result of the paper.
I found the most impressive aspect of this paper to be the fact that authors managed to shave the log factor in the rate (compared to the down-sampled version commonly used in Markovian setting).

**Strengths:**

The result of this paper seems rather satisfactory and tells a relatively complete story.
The main text is somewhat dense but it's never getting obscure.
Contributions, techniques to achieve them and difficulties for doing so are clearly explained.

**Weaknesses:**

I don't see obvious weaknesses.
So I'll just propose a few minor suggestions.

I personally found it useful to keep the following binary Markov chain in mind when reading the results and proof sketches: the states are -1, +1 and the transition kernel is given by a binary symmetric channel with flip probability delta.
Then the spectral gap is delta (perhaps up to constant) and the mixing time is essentially 1/delta.
I encourage the authors to use this as a running example in the proof sketch, though I don't strongly insist on this.

If I understand correctly, the Markov chain does not have to be finite-state right?
If so, this is perhaps worth commenting on, if it has not been done yet.

**Questions:**

1. The sin^2 guarantees in various theorem statements are somewhat heavy. Does it make sense to the authors to state a more compact bound (say using big-O notation) in the main text and defer the exact bound to appendices? On the other hand, I do understand that the authors would like to present completely nonasymptotic bounds...

2. I did not check the proof carefully, but my impression upon scanning the appendices is that the proof seems to rely heavily on the simplicity of the form of the Oja's iteration.
For instance, there're intimate relation between this work and a line of work on concentration for random matrix products.
Going beyond Oja's iteration, it possible to extrapolate the proof techniques to other iterations where there are nonlinearities?
E.g., in generalized power iteration, approximate message passing and so on, it's important to pick judicious nontrivial nonlinearities.
Of course, not all of these algorithms are applicable in the Markovian setting, but I'm sure the authors are aware of applicable examples.

3. The authors have already discussed the down-sampling trick in details. I would like to add a few more examples where there's loss of log factors using down-sampling. Bresler and his students have a line of work on inference in the presence of temporal correlation, notably this one https://arxiv.org/abs/2006.08916 in the context of linear regression. The idea there is precisely down-sampling and they showed down-sampling is optimal *up to log factors*. Clearly the authors of this submission is not satisfied with log factors. But it's still interesting to see similar phenomenon being recurrently observed.
Another "example" is in the setting of parameter estimation from Gaussian mixtures where the mixing coefficients form a binary Markov chain; see https://arxiv.org/abs/2206.02455.
Over there, curiously, one cannot waste samples. Instead, one takes coherent combination within blocks of size being roughly the mixing time of the chain.

---

> ### Author Rebuttal · Authors · 2023-08-09
>
> Thank you for your encouraging words about removing the logarithmic dependence in our rate.
>
> 1. *Regarding the example Markov chain*: Thank you for this suggestion. Due to the interpretable relationship between the spectral gap and the mixing time, we do use a similar Markov chain for our experimental setup (See lines 302-308). For added clarity, as per your suggestion, we will also try to include this in the proof sketch as a running example.
>
> 2. *Regarding finite state-space Markov chains*: We believe that our analysis should extend to the general state-space Markov chains, similar to the extensions in Section 4 in [A]. However, for ease of exposition, we have focused on the finite-state-space case in this paper.
>
> 3. Questions
>
>     Q1. *Regarding the exposition of theoretical guarantees*: We will add another version of the statement of Theorem 1 with reasonable values substituted for the learning rate. We do believe this will help the exposition.
>
>     Q2. *Regarding extrapolation to nonlinear updates*: We are currently working on the concentration of matrix products in the Markovian setting, which is an open problem to the best of our knowledge. Even in the IID case, non-linearities in the update may make the analysis deviate significantly from the Oja-type update, as in the case of streaming sparse PCA (see [B]).
>
>     Q3. *Regarding loss of log factors using down-sampling*: We are grateful to the reviewer for pointing out these references. We tried implementing the setting of Bressler et al.’s work [25] without discarding samples along with a decaying step size. Indeed, the algorithm that looked at the entire dataset performed better. Therefore, we believe this is a common phenomenon across multiple settings. We were unaware of the second paper and will cite it in our manuscript.
>
> References
>
> [A]. Lezaud, Pascal. Chernoff-type bound for finite Markov chains. Annals of Applied Probability (1998): 849-867.
>
> [B]. Yang, W. and Xu, H., 2015, June. Streaming sparse principal component analysis. In International Conference on Machine Learning (pp. 494-503). PMLR.

---

### Official Review · Reviewer_M3wf · 2023-07-07

**Soundness:** 3 good
**Presentation:** 3 good
**Contribution:** 3 good
**Rating:** 7
**Confidence:** 3

**Summary:**

This paper analyzes the performance of Oja'a algorithm for streaming PCA when the samples come from an irreducible, aperiodic, and reversible Markov chain. In this work the authors avoid using downsampling to reduce dependency, and their result improves on that proposed in [3] by a poly-logarithmic multiplicative factor.

**Strengths:**

This paper gives a near optimal upper bound on the estimation error of Oja's algorithm for streaming PCA with Markovian data. Most past results consider iid data, while in many practical applications the data are dependent. Their result improve upon previous error rate bounds, and nearly matches an offline algorithm. Careful comparisons of the error rates are also provided in the paper. Leveraging the mixing properties of Markov chains, they present novel analysis that might find applications in other online problems.

**Weaknesses:**

It seems that knowledge of the transition matrix is required to produce step sizes. In practice this must be estimated, it will be nice to include this as a part of the algorithm.

**Questions:**

1. What is C in line 128? C also appear as a constant later.
2. $\pi$ is the stationary distribution of $X$ or $s$? In line 46 $\pi$ seems to be the stationary distribution for $X$, while in section 2 $\pi$ seems to be the stationary distribution for $s$.
3. Assumption is not satisfied in several basic settings for example $X_i$ are Gaussian, which is a little disappointing. Can this assumption be relaxed using some additional arguments, for example, truncation?

**Limitations:**

Yes. Societal impact not applicable.

---

> ### Author Rebuttal · Authors · 2023-08-09
>
> Thank you for your kind words regarding the novelty of our analyses and wider applications for online problems.
>
> 1. *Regarding the estimation of the mixing time*: [A] provides an offline algorithm for estimating the mixing time of Markov chains. They show that at least a constant times  $\frac{\tau_{\text{mix}}|\Omega|}{\epsilon^{2}}$ samples are needed for estimating $\tau_{\text{mix}}$ within an absolute error $\epsilon$, where $|\Omega|$ denotes the size of the state space. Therefore, it is possible to use $O\left(\frac{1}{\epsilon^{2}}\right)$ samples to estimate the mixing time within accuracy $\epsilon$. We will add this as a remark in the manuscript.
>
>
>
>
> 2. Questions-
>
>     Q1. *Regarding $C$ on line 128*: $C$ here denotes the Markov Chain. We will remove this to avoid confusion.
>
>     Q2. *Regarding $\pi$ being the stationary distribution of $s$ or $X$*: With $E_{\pi}$, we meant the expectation over the stationary distribution of the states. We will replace this notation with $E_{s \sim \pi}E_{D(s)}\left[.\right]$ (the notation on line 140).
>
>     Q3. *Regarding truncation*: Thank you for asking this question. Indeed, we can extend our results to sub-gaussian data distributions with appropriate variance parameter decay. For space constraints and ease of exposition, we will provide an easy argument that changes the algorithm by considering "truncated" datapoints $\\{Y_i:= X_i1(\lVert X_i \rVert ^2\leq \alpha_n), i=1,\dots, n\\}$, where the truncation parameter $\alpha_n$ is set to be $c\log n$. Thus we replace any datapoint whose squared norm exceeds the truncation parameter by zero. We provide an argument to show that the principal eigenvector of the covariance matrix of this truncated distribution is sufficiently close to that of the original one. Standard arguments can then show that the sin-squared error of the output of this algorithm w.r.t the original principal eigenvector has the same theoretical guarantee.
>
>     To be concrete, for each state, consider a sub-gaussian distribution, where $\lVert X_{ij} \rVert_{\psi_2} \leq \nu_j$, and $\sum_j\nu_j<C$. Here $X_{ij}$ is the $j^{th}$ entry of random vector $X_i$, and $\lVert.\rVert_{\psi_2}$ is the sub-Gaussian Orlicz norm (see [D]). A similar setup for unbounded distributions has been used in the IID setting by [B] and [C].
>
>     We can show that this constrained distribution's covariance matrix $\Sigma' := \mathbb{E}\left[Y_{i}Y_{i}^{T}\right]$ is close in Frobenius norm to the original covariance matrix $\Sigma = \mathbb{E}\left[X_{i}X_{i}^{T}\right]$.
>
>     Next,  the Davis-Kahan theorem ([E]) can be applied to show that the population eigenvectors of $\Sigma$ and $\Sigma'$ are within a small sin-squared error, which can be made to be smaller than $1/n^b$ for a large $b$ (depending on the choice of $\alpha_n$). Thus the sin-squared error of this algorithm is still $O\left(\frac{\mathcal{V}}{n(1-|\lambda_2\left(P\right)|)(\lambda_1-\lambda_2)^2}\right)$ with high probability.  This is because the norm of the vectors (which is now $\Theta(\sqrt{\log n})$) only appears in the higher-order error terms. We also believe that it is possible to analyze the original algorithm directly by looking at the intersection of any relevant event with the set $\mathcal{A}_n=\\{\max_i \|X_i\|^2\leq \alpha_n\\}$. Proposition 1 of [B] shows that $P(\mathcal{A}_n)=1-o(1)$. We defer this argument for clarity.
>
> References
>
> [A] Wolfer, Geoffrey, and Aryeh Kontorovich. Estimating the mixing time of ergodic Markov chains. Conference on Learning Theory. PMLR, 2019.
>
> [B] Lunde, R., Sarkar, P. and Ward, R., 2021. Bootstrapping the error of Oja's algorithm. Advances in Neural Information Processing Systems, 34, pp.6240-6252.
>
> [C] Liang, X., 2023. On the optimality of the Oja's algorithm for online PCA. Statistics and Computing, 33(3), p.62.
>
> [D] R. Vershynin. High-Dimensional Probability. Cambridge University Press, Cambridge, UK,
> 2018.
>
> [E]. Yu, Yi, Tengyao Wang, and Richard J. Samworth. A useful variant of the Davis–Kahan theorem for statisticians. Biometrika 102.2 (2015): 315-323.

---

> > ### Comment · Reviewer_M3wf · 2023-08-17
> >
> > I want to thank the authors for the detailed response! My questions are well addressed, and I shall keep my current evaluation.

---

### Author Rebuttal · Authors · 2023-08-09

We want to first thank all the reviewers for their valuable suggestions and insightful feedback. We believe we have addressed nearly all of their main technical questions. We give brief sketches to show how to extend our analysis to **non-reversible** Markov chains, the **non-zero mean** data settings, and **sub-gaussian data distributions**.  In what follows, we will address some important points each reviewer has raised. We will correct all the typographical issues pointed out and will not address them here.

1. *Regarding reversibility (Reviewer eXbt)*:
We assumed reversibility to simplify our analyses. However, one can extend our results to the non-reversible case following reversibilization tools (also see [I] and [II]).  We use reversibility only in Lemma 3, which is then used by all other theorems. Here we look at the time-reversed chain, which is identical to the original Markov chain in the reversible case. The probability transition matrix $P_{\*}$ of the time-reversed Markov chain is given by $P_{\*}(x,y) = \frac{\pi(y)}{\pi(x)}P(y,x)$. In the non-reversible case, we consider the matrix $R_{\*}:=P_{\*}P$, which is also known as the multiplicative reversibilization of $P$ (see [II]). It corresponds to the transition matrix of a reversible Markov chain and has real and nonnegative eigenvalues. This allows us to replace the eigengap of the transition matrix $P$, i.e., $1-|\lambda_2(P)|$ in our current bound by $1-\sqrt{|\lambda_2(P_{\*}P)|}$. In our response to Reviewer eXbt, we provide more details.

2. *Regarding non-zero mean data (Reviewer iipj)*: Like the IID case, we can also deal with non-zero mean data by taking the difference of disjoint pairs of datapoints as our new data. This works for the streaming Markovian setting without increasing the mixing time. In our response to Reviewer iipj, we show that this essentially reduces to a modified Markov chain, which shares many important properties of the original Markov chain. Define the modified mean zero dataset as $\\{Y_i , i = 1,\dots,\lfloor \frac{n}{2} \rfloor \\}$ where $Y_{i} := X_{2i-1} - X_{2i}$. It is not hard to argue that this dataset is coming from another Markov chain with states $(a,b)$ such that $P(a,b)>0$ in the original Markov chain. We will denote the transition matrix of this Markov chain by $P'$, which is defined as follows: $P'((a,b),(c,d))=P(b,c)P(c,d) $. This is a valid transition matrix, and when the original Markov chain is irreducible and aperiodic, so is this one. For an argument on why the mixing time stays unchanged, please see our response to Reviewer iipj.

3. *Regarding sub-gaussian data distributions (Reviewer M3wf)*:
 One can extend our results to sub-gaussian data distributions for each state using a truncation-based argument as the reviewer points out. We consider sub-gaussian distributions with appropriate variance parameter decay similar to the setup used in the IID setting by [IV] and [V]. In our response to Reviewer M3wf, we show a roadmap to an argument, which makes a small modification to the algorithm by considering $\\{X_i1(\lVert X_i \rVert ^2\leq \alpha_n), i=1,\dots, n\\}$, where the truncation parameter $\alpha_n$ is set to be $c\log n$. Proposition 1 of [IV] shows that the probability of the largest squared norm of $n$ sub-gaussian vectors exceeding $\alpha_n$ is $o(1)$. Using this, we can extend our method to incorporate sub-gaussian distributions because the truncation parameter with logarithmic dependence on $n$ only appears in the higher-order error terms.

4. *Regarding estimation of mixing time (Reviewers M3wf, iipj)*:  [III] provides an offline algorithm for estimating the mixing time of Markov chains. They show that at least a constant times  $\frac{\tau_{\text{mix}}|\Omega|}{\epsilon^{2}}$ samples are need for estimation of the $\tau_{\text{mix}}$ within an absolute error $\epsilon$, where $|\Omega|$ denotes the size of the state space. Therefore, it is possible to use $O\left(\frac{1}{\epsilon^{2}}\right)$ samples to estimate the mixing time within accuracy $\epsilon$. We will add this as a remark in the manuscript.

5. *Regarding burn-in (Reviewer eXbt)*:  Bressler et al. [25] and Chen et al. [3], etc. also make the stationarity assumption when analyzing Markovian data. Please see the attached pdf for experiments observing the effect of different burn-in periods when starting from non-stationary distributions. We observe that the longer burn-in periods lead to a slightly faster error decay.

6. *Regarding presentation of Theorem 1 (Reviewers 1vdj, iipj, eXbt)*: We will add a simpler version of Theorem 1 with reasonable values substituted for the parameters for ease of exposition.

References

[I] Lezaud, Pascal. Chernoff-type bound for finite Markov chains. Annals of Applied Probability (1998): 849-867.

[II] J. A. Fill. Eigenvalue bounds on convergence to stationarity for non-reversible Markov chains,
with an application to the exclusion process. The Annals of applied probability pages 62–87,
1991.

[III] Wolfer, Geoffrey, and Aryeh Kontorovich. Estimating the mixing time of ergodic Markov chains. Conference on Learning Theory. PMLR, 2019.

[IV] Lunde, R., Sarkar, P. and Ward, R., 2021. Bootstrapping the error of Oja's algorithm. Advances in Neural Information Processing Systems, 34, pp.6240-6252.

[V] Liang, X., 2023. On the optimality of the Oja’s algorithm for online PCA. Statistics and Computing, 33(3), p.62.

---

### Decision · Program_Chairs · 2023-09-21

**Decision:**

Accept (spotlight)

**Comment:**

All reviewers appreciate the important problem of analyzing algorithms in the Markovian setting as opposed to iid data. The paper gives a nearly tight bound without losing a log factor for Oja's algorithm. This log factor is common in bounds for many algorithms in the Markovian setting due to a simpler analysis using down-sampling. On the downside, the paper needs to improve the presentation, which the authors promised in the rebuttals.